# Impact of the Coronavirus Disease-2019 Pandemic on Pancreaticobiliary Disease Detection and Treatment

**DOI:** 10.3390/jcm10184177

**Published:** 2021-09-16

**Authors:** Muneo Ikemura, Ko Tomishima, Mako Ushio, Sho Takahashi, Wataru Yamagata, Yusuke Takasaki, Akinori Suzuki, Koichi Ito, Kazushige Ochiai, Shigeto Ishii, Hiroaki Saito, Toshio Fujisawa, Akihito Nagahara, Hiroyuki Isayama

**Affiliations:** Department of Gastroenterology, Graduate School of Medicine, Juntendo University, 2-1-1 Hongo, Bunkyo-ku, Tokyo 113-8421, Japan; m-ikemura@juntendo.ac.jp (M.I.); tomishim@juntendo.ac.jp (K.T.); m-ushio@juntendo.ac.jp (M.U.); sho-takahashi@juntendo.ac.jp (S.T.); w.yamagata.mx@juntendo.ac.jp (W.Y.); ytakasa@juntendo.ac.jp (Y.T.); suzukia@juntendo.ac.jp (A.S.); kitoh@juntendo.ac.jp (K.I.); k.ochiai.qd@juntendo.ac.jp (K.O.); sishii@juntendo.ac.jp (S.I.); hiloaki@juntendo.ac.jp (H.S.); t-fujisawa@juntendo.ac.jp (T.F.); nagahara@juntendo.ac.jp (A.N.)

**Keywords:** COVID-19, ERCP, EUS, pancreaticobiliary disease, pandemic, SARS-CoV-2

## Abstract

The emergency declaration (ED) associated with the coronavirus disease-2019 (COVID-19) pandemic in Japan had a major effect on the management of gastrointestinal endoscopy. We retrospectively compared the number of pancreaticobiliary endoscopies and newly diagnosed pancreaticobiliary cancers before (1 April 2018 to 6 April 2020), during (7 April to 25 May 2020), and after the ED (26 May to 31 July). Multiple comparisons of the three groups were performed with respect to the presence or absence of symptoms and clinical disease stage. There were no significant differences among the three groups (Before/During/After the ED) in the mean number of diagnoses of pancreatic cancer and biliary cancer per month in each period (8.0/7.5/7.5 cases, *p* = 0.5, and 4.0/3.5/3.0 cases, *p* = 0.9, respectively). There were no significant differences among the three groups in the number of pancreaticobiliary endoscopies (EUS: endoscopic ultrasonography/ERCP: endoscopic retrograde cholangiopancreatography) per month (67.8/62.5/69.0 cases, *p* = 0.7 and 89.8/51.5/86.0 cases, *p* = 0.06, respectively), whereas the number of EUS cases decreased by 42.7% between before and during the ED. There were no significant differences among the three groups in the presence or absence of symptoms at diagnosis or clinical disease stage. There was no significant reduction in the newly diagnosed pancreaticobiliary cancer, even during the ED. The number of ERCP cases was not significantly reduced as a result of urgent procedures, but the number of EUS cases was significantly reduced.

## 1. Introduction

COVID-19 is an infectious disease that can lead to serious respiratory disorders caused by severe acute respiratory syndrome coronavirus-2 (SARS-CoV-2) and which spread worldwide from China (Wuhan) around November 2019, causing a pandemic. SARS-CoV-2 is transmitted through droplet infection and contact infection, but airborne transmission via aerosolized SARS-CoV-2 is also a concern [1,2]. In Japan, infection spread from around January 2020, and an emergency declaration (ED) was issued for the period between 7 April 2020 and 25 May 2020. To prevent the spread of infection, the following was requested: refraining from going outside, closing of schools, and restrictions on the use of facilities where many people gather, such as department stores and movie theaters.

During gastroenterological endoscopy, an aerosol generated by coughing during endoscope insertion is a risk because of the proximity of medical workers and patients. SARS-CoV-2 can survive for several hours in the air and may be transmitted by prolonged exposure to high concentrations of contaminated aerosols in enclosed spaces, such as the endoscope room [3]. Additionally, there is potential for fecal virus shedding [4,5], which is a potential risk of infection in colonoscopy. Therefore, the number of endoscopies for gastrointestinal screening has decreased, possibly delaying detection of gastrointestinal cancer [6]. Emergency endoscopic procedures should be performed under strict infection control. Especially in the field of pancreaticobiliary disease, such as severe cholangitis, delay is dangerous, and it is necessary to commence diagnosis and treatment promptly, considering the poor prognosis of pancreaticobiliary cancer. Pancreaticobiliary cancer is often at an advanced stage at diagnosis, which contributes to the poor prognosis. Because endoscopic treatment for pancreaticobiliary cancer is usually in a symptomatic context, it is important to examine the changes in endoscopic examination of the bile and pancreas under COVID-19 to evaluate the usefulness of screening for asymptomatic pancreaticobiliary cancer.

We compared the number of cases of pancreaticobiliary endoscopy before and after the ED and the number of newly diagnosed pancreaticobiliary cancers, and examined the period-by-period correlation between the presence of symptoms and tumor progression.

## 2. Methods

### 2.1. Study Design

This was a single-center retrospective study and was approved by our institutional review board (ethical code 20-232). We reviewed the chart and database of endoscopy and radiological procedures. All authors had access to the study data and approved the final manuscript.

We examined the number of endoscopic treatments for pancreaticobiliary disease (EUS: endoscopic ultrasonography/ERCP: endoscopic retrograde cholangiopancreatography) and newly diagnosed pancreaticobiliary cancer (BTC: biliary tract cancer/PC: pancreatic cancer) in our hospital. We analyzed BTC including intrahepatic bile duct cancer, hilar bile duct cancer, distal bile duct cancer, gallbladder cancer, and papilla of vater cancer. We surveyed three periods: before (1 April 2018 to 6 April 2020), during (7 April 2020 to 25 May), and after the ED (26 May to 31 July). We retrospectively compared the age, sex, clinical stage, and symptoms of patients with pancreaticobiliary cancer diagnosed during each period. Symptoms were defined as jaundice, fever, abdominal pain, and weight loss.

### 2.2. Infection Protection Measures of Endoscopic for Coronavirus in Our Hospital

During the COVID-19 period, patients were required to wear masks with face-shields (or goggles and masks), gloves, caps, and gowns (long sleeves) when undergoing endoscopy. After the examination and treatment were completed, the fingers and elbows were cleaned thoroughly and disinfected with alcohol. All procedures were performed with medical staff wearing N95 masks and surgical masks. For high-risk patients (positive PCR or antigen test; persistent fever and/or dyspnea; computed tomography (CT) findings of pneumonia, dysgeusia, and/or dysosmia; and close contact with COVID-19 patients within 2 weeks), endoscopic treatment was performed in a continuous negative pressure room with double-layer gloves, covering on the whole body, and ventilation as recommended by the Ministry of Health, Labor, and Welfare (30 m^3^ per person per hour) [7,8]. The minimum number of endoscopists, assistants, and nurses were used for the procedure.

### 2.3. Statistical Analysis

The number of endoscopic procedures (ERCP/EUS) and the number of diagnoses of pancreaticobiliary cancer (PC, BTC) and the presence or absence of symptoms, and the proportion of stage III/IV at diagnosis, were compared via Kruskal–Wallis test and the Steel–Dwass method during the three periods. The Mann–Whitney *U*-test was used for two-group comparisons. The α-level was defined as 0.05, and probability values less than the α-level were considered to be statistically significant.

## 3. Results

### 3.1. Details of Newly Diagnosed Pancreaticobiliary Cancer at Each Period

The mean number of diagnoses per month before, during, and after the ED was 8.0, 7.5, and 7.5 cases, respectively, with no significant difference among the three groups (*p* = 0.5). The mean age at the time of pancreatic cancer diagnosis was 69.6, 69.7, and 71.6 years, and the proportion of males was 51.5%, 53.3%, and 55.6%, with no significant difference among the three groups. The presence of symptoms at diagnosis and the clinical stage of cancer (rate of stage III and IV) were more in evidence and more advanced during the ED, but there was no significant difference among the three groups (77.5%/93.3%/80.0%, *p* = 0.3 and 72.5%/80.0%/60.0%, *p* = 0.9, respectively) (Table 1).

The mean number of diagnoses of biliary tract cancer per month was 4.0, 3.5, and 3.0, and there was no significant difference among the three groups (*p* = 0.9). The mean age at the time of diagnosis of biliary tract cancer was 71.8, 71.4, and 71.8, and the rate of males was 67.3%, 71.4%, and 66.7%, showing no significant difference among the three groups. The rate of symptoms was 82.5%, 57.1%, and 100%, which tended to be slightly but non-significantly lower during the ED, but there was no significant difference (*p* = 0.4). There was no significant difference in stage III and IV disease (62.0%/83.0%/88.0%) (*p* = 0.5) (Table 1).

### 3.2. The Average Number of Biliary and Pancreatic Endoscopes at Each Period (Monthly Average)

The number of biliary and pancreatic endoscopies decreased. There was no significant difference in the number of pancreaticobiliary endoscopies performed per month among the three groups (67.8/62.5/69.0 cases), whereas the number of EUSs was 89.8, 51.5, and 86.0, a 42.7% decrease between before and during the ED. Multiple comparisons showed a significant reduction between before and during the ED (*p* = 0.04, Figure 1). The number of EUS-guided fine needle aspiration (EUS-FNA) cases also tended to be lower (17.8, 11.5, and 18.5 cases) during the ED, but the difference was not significant (*p* = 0.7) (Table 2).

### 3.3. The Breakdown of ERCP during and Post Emergency Declaration (Monthly Average)

The ERCP breakdown between during and after the ED showed no change in the treatment of malignant tumor, benign biliary tract workup, or stone. Regarding cholecystitis, one patient was treated percutaneously during the ED. Endoscopic transpapillary gallbladder drainage (ETGBD) was enforced for four cases of cholecystitis after the ED (Table 3).

## 4. Discussion

All upper gastrointestinal endoscopies (including ERCP/EUS) have a risk of aerosol generation. To prevent aerosols caused by the cough reflex, sedation is recommended. For pancreaticobiliary endoscopic procedures such as EUS and ERCP, the procedure duration is longer than other upper endoscopic procedures. Caution is also needed with regards to aerosols when placing and removing a number of devices in and from the working channel. In these aspects, pancreaticobiliary endoscopic procedures may be associated with a higher risk than upper gastrointestinal endoscopies. At the time of the emergency declaration, it was recommended that the decision for examination and treatment of each case should take into account the extent of COVID-19 infection, the triage and risk of infection by case, and the circumstances of the hospital setting [9].

In this study, we investigated the impact of coronavirus epidemics on the diagnosis of malignancy using the number of endoscopic examinations and the number of newly diagnosed pancreaticobiliary cancer our hospital. The results showed no significant difference in the number of diagnoses of pancreatic cancer and biliary tract cancer before, during, and after the ED. The functioning of the hospital and the maintenance of a similar diagnosis rate for pancreaticobiliary cancer to that of before the ED, without a cluster, were considered to be due to adequate triage and protection against infection. However, more than 60% of the cases were stage III or higher at diagnosis. This underscores the importance of screening for pancreaticobiliary cancer, given that early diagnosis before symptom onset was not possible. Gastrointestinal cancer can be detected early by screening endoscopy; therefore, a reduction in the rate of screening endoscopy will reduce that of gastrointestinal cancer detection. The number of pancreaticobiliary endoscopies decreased. The rate of EUS was significantly reduced, by 42.7%, and that of EUS-FNA by 23.3% between during and before the ED (Figure 2). This may be due to a shift from EUS to magnetic resonance cholangiopancreatography (MRCP) follow-up, especially in intraductal papillary mucinous neoplasm (IPMN) without high-risk symptoms and asymptomatic choledocholithiasis [10]. Priority was given to symptomatic patients, patients with masses on CT or MRI/MRCP, and those at high suspicion of malignancy [8,11]. Pancreatic cancer should be diagnosed and treated aggressively because of its rapid progression and limited resectability. The European Society of Medical Oncology (ESMO) recommends aggressive workup of pancreatic cancer in patients with suspected cancer on imaging, jaundice, or gastrointestinal obstruction [12]. Although it did not influence the diagnosis rate or stage of malignancy, further studies are needed to determine the magnitude of the delay in the diagnosis of cancer caused by a reduced rate of pancreaticobiliary EUS screening. Regarding the ERCP breakdown, there were no differences between during and after the ED in stent occlusion (benign/malignancy) or symptomatic bile duct stones. Endoscopic drainage (ETGBD) is the first-line modality for cholecystitis, but during the ED, patients were switched to percutaneous transhepatic gallbladder drainage (PTGBD) based on the risk of aerosol generation during endoscopy. There were no clinical problems with switching to PTGBD.

Emergency cases in gastroenterology include gastrointestinal bleeding, obstructive jaundice, biliary tract infection, acute pancreatitis, appendicitis, strangulating ileus, and acute large bowel obstruction. If the patient has a fever, we should be sufficiently cautious about delaying treatment of emergency cases. However, if consultations with a specialist are delayed until after a negative COVID-19 test result, it may be too late for theses emergency cases. In our hospital, EUS-hepaticogastrostomy (EUS-HGS) stenting was performed to prevent recurrence of bile duct cancer after surgery, and if the patient visited our hospital because of cholangitis. Because pneumonia was recognized by CT, a patient was put under 1-day observation and an antibiotic agent prescribed until a negative PCR result ensued. However, on day 2, the delayed treatment led to multiple organ failure. Thus, it is important not to miss the window for treatment of these emergency diseases. Especially, patients with malignancies have less spare ability than healthy individuals and are more likely to have severe infections such as cholangitis. In such situations, ERCP, including elective stent replacement procedures, should be performed without delay [13]. Hepatic disorders have been noted in about 20% of patients with COVID-19 symptoms, and differentiation from cholangitis may be difficult in patients with fever and hepatic dysfunction [14,15]. Elevated pancreatic enzymes were reported in 17% of patients [12]. High amylase without abdominal pain is common, and attention must be paid to pancreatic enzymes such as amylase and lipase when examining patients with COVID-19. CT of the abdomen is necessary to rule out pancreatitis. Thus, with regards to pancreaticobiliary disease and COVID-19, it is important to determine whether the disease is aggravated or a side effect of COVID-19, and to perform necessary treatment without delay [12,13,14,15].

### Limitation

Simple comparisons are difficult because of the different lengths and timing of the three periods compared. It is a limitation that the number of cases was small in the examination of a single department.

## 5. Conclusions

The diagnosis of pancreaticobiliary cancer should be made with adequate protection against infection in COVID-19. In addition, treatment with symptoms such as cholangitis was appropriately performed during the ED. On the other hand, the number of asymptomatic screening procedures has decreased, and the impact on the early diagnosis of pancreaticobiliary cancer needs further investigation.

## Figures and Tables

**Figure 1 jcm-10-04177-f001:**
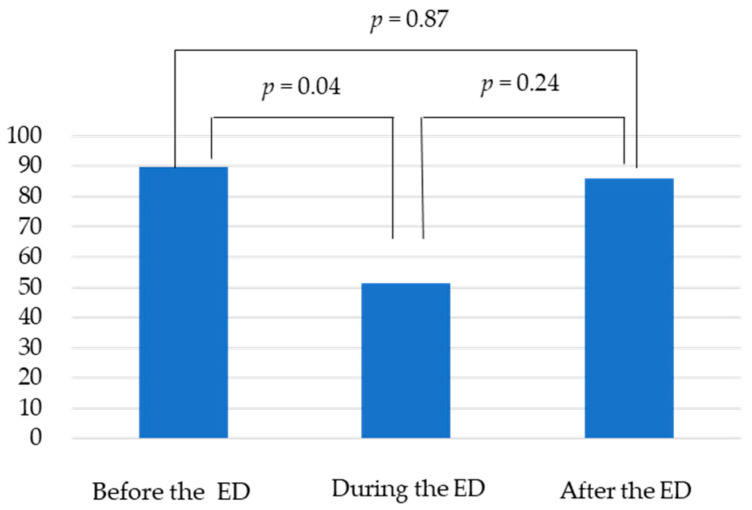
There were no significant differences among the three groups in the number of EUS per month (89.8/51.5/86.0 cases). Multiple comparisons showed a significant reduction between before and during the ED (*p* = 0.04).

**Figure 2 jcm-10-04177-f002:**
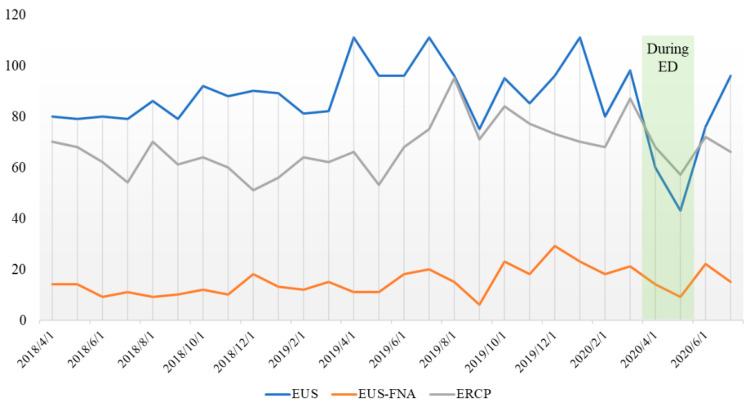
The number of pancreaticobiliary endoscopies decreased. The rate of EUS was significantly reduced by 42.7%, and that of EUS-FNA was reduced by 23.3% between during and before the ED.

**Table 1 jcm-10-04177-t001:** Details of pancreaticobiliary Cancer at Each Period.

	Before the ED	During the ED	After the ED	*p* Value
Duration	1 April 2018–6 April 2020	7 April 2020–25 May 2020	26 May 2020–31 July 2020	
PC cases, total	96	15	15	
PC cases, average/m	8.0	7.5	7.5	0.5
Age average	69.6	69.7	71.6	0.3
Sex (Male), %	51.5	53.3	55.6	0.4
Symptomatic case, %	77.5	93.3	80	0.3
Stage III/IV, %	72.5	80	60	0.9
BTC cases, total	48	7	6	
BTC cases, average/m	4.0	3.5	3.0	0.9
Age average	71.8	71.4	71.8	0.3
Sex (Male), %	67.3	71.4	66.7	0.3
Symptomatic case, %	82.5	57.1	100	0.4
Stage III/IV, %	62	83	88	0.5

ED: emergency declaration, BTC: biliary tract cancer, PC: pancreatic cancer.

**Table 2 jcm-10-04177-t002:** The number of biliary and pancreatic endoscopes at each period (monthly average).

	Before the ED	During the ED	After the ED	*p* Value
Duration	1 April 2018–6 April 2020	7 April 2020–25 May 2020	26 May 2020–31 July 2020	
EUS	89.8	51.5	86.0	0.06
EUS-FNA	15.0	11.5	18.5	0.3
ERCP	67.8	62.5	69.0	0.7

EUS: endoscopic ultrasounds, EUS-FNA: EUS-guided fine needle aspiration, ERCP: endoscopic retrograde cholangiopancreatography.

**Table 3 jcm-10-04177-t003:** The breakdown of ERCP during and post emergency declaration (monthly average).

	During the ED	After the ED	*p* Value
Duration	7 April 2020–25 May 2020	26 May 2020–31 July 2020	
Malignant stricture			
Diagnosis	15.0	15.0	0.2
Exchange stent	10.0	9.0	0.6
Benign stricture			
Diagnosis	1.0	5.0	0.3
Exchange stent	19.0	16.5	1.0
CBDS	11.5	15.0	0.2
ETGBD	0.5	2.0	0.6

CBDS: common bile duct stones, ETGBD: endoscopic transpapillary gallbladder drainage.

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
