# Peer review of "Impact of the Coronavirus Disease-2019 Pandemic on Pancreaticobiliary Disease Detection and Treatment"

_jcm, 2021, doi:10.3390/jcm10184177_

Round 1
Reviewer 1 Report
Ikemura M, et al. examined a serial change in the number of pancreatobiliary endoscopy procedures during the COVID-19 pandemic in Japan. The authors found a decrease in diagnostic EUS during the state of emergency in 2020. Overall, this study was well designed with clinical relevance in the era of COVID-19, and I only have some minor comments.
#1. In the Abstract section and other corresponding parts, the authors should state explicitly that they discussed the numbers of “newly-diagnosed” pancreatobiliary cancer.
#2. The alpha level used for statistical significance should be described.
Author Response
Sep 8, 2021
Journal of Clinical Medicine
Dear Assigned Editor prof. Yvette Li and Reviewers
Thank you very much for reviewing our manuscript and offering valuable advice. We have addressed your comments with point-by-point responses, and revised the manuscript accordingly. Please find the revised version of the manuscript entitled “jcm-1351937. Impact of the Coronavirus Disease-2019 Pandemic on Pancreaticobiliary Disease Detection and Treatment” with tables and figures to be considered for publication in JCM.
Please contact me if there are further questions regarding this revised manuscript. We appreciate if decision of acceptance on this manuscript would be transferred by e-mail. Thank you for your consideration. We are looking forward to hearing from you.
Muneo Ikemura, M.D.
Hiroyuki Isayama, M.D., Ph.D.
Department of Gastroenterology, Juntendo University, School of Medicine,
2-1-1, Hongo, Bunkyo-ku, Tokyo, 113-8421, Japan
Phone; +81-3-5802-1060
Fax; +81-3-3813-8862
E-mail; tomishim@juntendo.ac.jp
Reviewer 1
Ikemura M, et al. examined a serial change in the number of pancreatobiliary endoscopy procedures during the COVID-19 pandemic in Japan. The authors found a decrease in diagnostic EUS during the state of emergency in 2020. Overall, this study was well designed with clinical relevance in the era of COVID-19, and I only have some minor comments.
#1. In the Abstract section and other corresponding parts, the authors should state explicitly that they discussed the numbers of “newly-diagnosed” pancreatobiliary cancer.
→Thank you for your great suggestion. We corrected it as follow; P1 Line13 and Line24 (abstract), P2 Line56 (intro), P2 Line66 (methods), P3 Line94 (Results) and P5 Line153 (Discussion).
#2. The alpha level used for statistical significance should be described.
→Thank you for your great suggestion. We added it as follow; “The α-level was defined as o.o5, and probability values less than the α-level are considered to be statistically significant.” (P2 Line 91-92)

Reviewer 2 Report
Please refer to attached pdf file of the manuscript with yellow highlights which denote where changes need to be made.
Most are typographical errors to be corrected. The exceptions are:
Line 187: "Especially, the cases of the malignant tumor tend to be serious, since the organ space ability is not left in the case." ; this sentence is not comprehensible, and needs re-writing
Line 204-205 in the Conclusion: the statement made there is quite discordant from the conclusion in the abstract. The key message in the abstract conclusion and main paper conclusion need to be harmonised.
Author Response
Sep 8, 2021
Journal of Clinical Medicine
Dear Assigned Editor prof. Yvette Li and Reviewers
Thank you very much for reviewing our manuscript and offering valuable advice. We have addressed your comments with point-by-point responses, and revised the manuscript accordingly. Please find the revised version of the manuscript entitled “jcm-1351937. Impact of the Coronavirus Disease-2019 Pandemic on Pancreaticobiliary Disease Detection and Treatment” with tables and figures to be considered for publication in JCM.
Please contact me if there are further questions regarding this revised manuscript. We appreciate if decision of acceptance on this manuscript would be transferred by e-mail. Thank you for your consideration. We are looking forward to hearing from you.
Muneo Ikemura, M.D.
Hiroyuki Isayama, M.D., Ph.D.
Department of Gastroenterology, Juntendo University, School of Medicine,
2-1-1, Hongo, Bunkyo-ku, Tokyo, 113-8421, Japan
Phone; +81-3-5802-1060
Fax; +81-3-3813-8862
E-mail; tomishim@juntendo.ac.jp
Reviewer 2
Please refer to attached pdf file of the manuscript with yellow highlights which denote where changes need to be made.
Most are typographical errors to be corrected. The exceptions are:
Line 187: "Especially, the cases of the malignant tumor tend to be serious, since the organ space ability is not left in the case." ; this sentence is not comprehensible, and needs re-writing
→Thank you for your comment. We corrected it as follow on P6 Line 193-194; “Especially, patients with malignancies have less spare ability than healthy individuals and are more likely to have severe infections such as cholangitis.”
Line 204-205 in the Conclusion: the statement made there is quite discordant from the conclusion in the abstract. The key message in the abstract conclusion and main paper conclusion need to be harmonised.
→Thank you for your valuable comment. We corrected it on P6 Line 209-213. “The diagnosis of pancreaticobiliary cancer should be made with adequate protection against infection in COVID-19. In addition, treatment with symptoms such as cholangitis was appropriately performed during the ED. On the other hand, the number of asymptomatic screening procedures has decreased, and the impact on the early diagnosis of pancreaticobiliary cancer needs further investigation.”
